# Anti-Tumor Potential of a 5-HT3 Receptor Antagonist as a Novel Autophagy Inducer in Lung Cancer: A Retrospective Clinical Study with In Vitro Confirmation

**DOI:** 10.3390/jcm8091380

**Published:** 2019-09-03

**Authors:** Jeong Soo Lee, Seong Yong Park, Na Young Kim, Dong Wook Kim, Ju Eun Oh, Eunjin Heo, Jong Seok Lee, Young Chul Yoo

**Affiliations:** 1Department of Anesthesiology and Pain Medicine, Yonsei Cancer Center, Severance Hospital, Yonsei University College of Medicine, Seoul 03722, Korea; RATION99@yuhs.ac (J.S.L.); KNNYYY@yuhs.ac (N.Y.K.); 2Anesthesia and Pain Research Institute, Yonsei University College of Medicine, Seoul 03722, Korea; 3Department of Thoracic and Cardiovascular Surgery, Severance Hospital, Yonsei University College of Medicine, Seoul 03722, Korea; SYPARKCS@yuhs.ac; 4Department of Policy Research Affairs, National Health Insurance Service Ilsan Hospital, Goyang 10444, Gyeonggi-do, Korea; kimdw2269@gmail.com; 5Department of Anesthesiology and Pain Medicine, Gangnam Severance Hospital, Yonsei University College of Medicine, Seoul 06273, Korea; EJHEO0919@yuhs.ac

**Keywords:** lung neoplasm, palonosetron, ramosetron, serotonin antagonist

## Abstract

Unlike 5-hydroxytryptamine (5-HT, serotonin) 1 and 5-HT2, the effect of 5-HT3 receptors on tumor cells is poorly understood. We conducted this study to determine whether the perioperative use of 5-HT3 receptor antagonists, which are widely used antiemetics, impacts the recurrence and mortality after lung cancer surgery and related anti-tumor mechanisms. From data on 411 patients, propensity score matching was used to produce 60 1:2 matched pairs of patients, and variables associated with the prognosis after open lung cancer surgery were analyzed. Additionally, the effects of 5-HT3 receptor antagonists were confirmed in vitro on A549 human lung adenocarcinoma cells. Cancer recurrence occurred in 10 (8.2%) and 14 (22.95%) patients (*p* = 0.005), treated or untreated, with palonosetron or ramosetron. Perioperative usage of palonosetron or ramosetron was also associated with lower recurrence rate after lung cancer surgery (hazard ratio (HR), 0.293; 95% confidence interval (CI) 0.110–0.780, *p* = 0.0141). Our in vitro experiments also showed that palonosetron and ramosetron inhibited cell proliferation and colony formation and reduced migration, which was associated with autophagic cell death via the extracellular signal-regulated kinase (ERK) pathway. Palonosetron and ramosetron may have anti-tumor potential against lung cancer cells, suggesting the need to consider these drugs as first-choice antiemetics in patients undergoing lung cancer surgery.

## 1. Introduction

Even though surgical resection is the first-choice treatment of solid tumors, tumor cells not visible to the eye can remain, seed, or migrate during surgery [1,2,3,4]. These remnant tumor cells can be removed by a patient’s own immune system [5]; however, patients undergoing cancer surgery are often older in age and immunocompromised [4,6,7]. Moreover, some anesthetics, as well as surgical stress, were shown to further compromise patient immunity perioperatively. All of this can affect mortality and the long-term prognosis of the patient [8,9,10,11]. Accordingly, efforts to prevent perioperative immunosuppression by mitigating increases in stress hormone and sympathetic activation and to choose anesthetic agents and other drugs for minimizing recurrence after cancer surgery garnered increasing interest in anesthesiology research.

Lung cancer is the leading cause of cancer deaths among men and women worldwide [12]. Even after surgical resection, which can only be performed in fewer than 20% of patients diagnosed with lung cancer, recurrence is frequent. In pathologic stage IA non-small-cell lung cancer, more than 10% of patients experience recurrence after surgery [13]. The frequency of recurrence and poor survival in lung cancer make it necessary to investigate remedial methods of lowering recurrence rates.

The antiemetic 5-hydroxytryptamine (5-HT, serotonin) 3 receptor antagonist (5-HT3RA) suppresses the action of serotonin, which is a neurotransmitter that sends signals to the gastrointestinal tract from the brain, resulting in vomiting [14]. Previous studies showed that serotonin acts as a mitogenic factor in both normal and cancer cells and promotes the proliferation of various types of cancer cells [15,16,17]. Most studies concerning the tumor mitogenic effects of serotonin focused on 5-HT1 and 5-HT2 receptors, but not 5-HT3 receptors [15,16,18,19].

We hypothesized that 5-HT3 receptors would also exhibit mitogenic potential and that post-operative administration of its antagonist would improve oncologic outcomes after surgery. In this study, we retrospectively investigated the effects of post-operative 5-HT3RA on recurrence and mortality in patients that underwent open lung cancer surgery. Also, an in vitro study was performed to identify the mechanism via which 5-HT3RA exerts its favorable effects on lung cancer cells.

## 2. Experimental Section

### 2.1. Retrospective Study

#### 2.1.1. Study Population and Design

This retrospective study was approved by the Institutional Review Board (IRB) of Severance Hospital, Yonsei University Health System, Seoul, South Korea (protocol number: 4–2018-0042; date of approval: 11 March 2018). Because this was a retrospective study using electronic medical records of anonymous patients who underwent lung cancer surgery, the IRB granted a waiver for individual consent.

Data were collected from patients who underwent open lung cancer surgery between January 2009 and December 2014 in a tertiary hospital in Seoul, South Korea. Data from patients who received anatomic resection, such as lobectomy and complete mediastinal lymph node dissection, were analyzed. Demographic data, including age, sex, height, weight, and co-morbidities, such as hypertension, diabetes mellitus, pulmonary diseases, were collected. In addition to cancer stage, pre-post-operative chemotherapy or radiation therapy, type of anesthetic, common intraoperative drugs, intraoperative colloid administration, and red blood cell transfusion were also analyzed as variables that could potentially be associated with the prognosis of lung cancer after open surgery. Lung cancer stage was measured according to the eighth edition of the American Joint Committee on Cancer staging system.

#### 2.1.2. Statistical Analysis

The primary objective of this study was to compare recurrence and mortality rates in lung cancer between perioperative palonosetron or ramosetron administration with and without 5-HT3RA. The maximal follow-up period was set at 60 months. Demographic and perioperative characteristics were analyzed using Student’s *t*-test for continuous variables and the chi-square test for categorical variables. Potential factors affecting cancer recurrence and mortality after surgery were analyzed using competing risk and the Cox proportional hazards model, and the risk of each variable was calculated as a hazard ratio (HR) and 95% confidence interval (CI). Fine and Gray competing risk analysis was performed for recurrence, with death as a competing risk. After screening for potentially significant variables in univariate analysis, multiple Cox regression analysis was performed including factors in univariate analysis with *p*-values < 0.2. Overall survival rates during the study period and the difference between survival curves according to palonosetron or ramosetron treatment were examined using the Kaplan–Meier log-rank survival analysis method. Propensity score matching (PSM) was applied to reduce selection bias, and confounders used for PSM included age, sex, and cancer stage. Taking into account the original proportion of patient-to-group numbers and the total sample size with statistical significance, 1:2 matching was conducted with propensity scores calculated with logistic regression analysis with the aforementioned confounding factors. The risk of recurrence and mortality with perioperative palonosetron or ramosetron administration with and without 5-HT3RA was analyzed with conditional Cox proportional hazard regression. A *p*-value less than 0.05 was considered as statistically significant. All statistical analyses were conducted with SAS version 9.4 (SAS Institute Inc., Cary, NC, USA) except for Kaplan–Meier curves, which were constructed using the R package version 3.0.2 (www.r-project.org).

### 2.2. In Vitro Study

#### 2.2.1. Cell Cultures and Drug Treatment

The A549 human lung adenocarcinoma cell line was purchased from American Type Culture Collection (Manassas, VA, USA). The cells were cultured in Roswell Park Memorial Institute (RPMI)-1640 medium (Hyclone, Logan, Utah, USA) supplemented with 10% heat-inactivated fetal bovine serum (FBS), l-glutamine, 100 IU/mL penicillin, and 100 μg/mL streptomycin at 37 °C in a humidified atmosphere of 5% CO_2_. Ondansetron, palonosetron, and ramosetron purchased from Sigma-Aldrich were freshly dissolved in RPMI-1640 medium and applied to the A549 cells for 48 h. Bafilomycin A1 (Baf-1), a potent inhibitor of cellular autophagy, was purchased from Sigma-Aldrich.

#### 2.2.2. Flow Cytometry

A549 cells were detached with 0.125% trypsin–ethylenediaminetetraacetic acid (EDTA) and fixed in 2% paraformaldehyde in phosphate-buffered saline (PBS) for 15 min at 4 °C. The fixed cells were then blocked in 10% FBS for 30 min on ice. Primary antibodies against 5-HT3A, 5-HT3B, 5-HT3C, and 5-HT3D + E (Bioss, Massachusetts, USA) and secondary Alexa Fluor 555 immunoglobulin G (Invitrogen) were added to the cell suspension and incubated at 4 °C in the dark for 1 h. The labeled cells were resuspended in PBS containing 2% FBS after washing with incubation buffer and analyzed using a BD FACScan flow cytometer (Becton Dickinson Biosciences).

#### 2.2.3. Cell Viability

A549 cells were grown in 96-well plates to a density of 5 × 10^4^ cells/mL before treatment with ondansetron, palonosetron, or ramosetron for 24 or 48 h in complete media. Cells treated with solvent only were used as a control. Cell viability was assessed using the EZ-Cytox Cell Viability Assay Kit (DOGEN, Seoul, South Korea). Absorbance at 450 nm was measured for the experimental groups using a plate reader. Experiments were performed in six biological replicates.

#### 2.2.4. Cell Scraping Assay

A549 cells were seeded onto six-well plates at a density of 1 × 10^5^ cells/mL and cultured in a humidified atmosphere containing 5% CO_2_ at 37 °C. After 24 h, the cells reached 80–90% confluence as a monolayer, which was gently scraped with a new 1-mL pipette tip across the center of each well. The well was then washed gently with PBS three times to remove detached cells. Remaining cells were grown in RPMI with 2% FBS containing ondansetron, palonosetron, or ramosetron for an additional two days, after which images were captured using a light microscope (×10 magnification). Then, the number of migrated cells was counted.

#### 2.2.5. Clonogenic Assay

A549 cells were seeded onto six-well plates at 5  × 10^4^ cells/well and treated with ondansetron, palonosetron, or ramosetron for two days. Next, 200 cells were plated per well into six-well plates and incubated for an additional 10 days. Images of colonies were captured using a light microscope (×10 magnification), and colony size was measured using Image J software (NIH, Bethesda, MD, USA).

#### 2.2.6. Western Blot Analysis

A549 cells were treated with ondansetron, palonosetron, or ramosetron for 48 h, washed twice with cold PBS, and lysed in radio-immunoprecipitation assay buffer containing protease inhibitors (10 µg/mL each of aprotinin, bestatin, l-leucine, and pepstatin A) dissolved in a solution of 150 mM NaCl, 10 mM Tris, 1 mM EDTA, 1 mM benzenesulfonyl fluoride, and 1% NP-40. Total protein concentration was determined by Quick Start Bradford reagent (Bio-Rad, Hercules, CA, USA). Whole-cell extracts (50 μg) were separated on 10–15% sodium dodecyl sulfate–polyacrylamide gel electrophoresis gels and transferred onto Immobilon P-transfer membranes (Millipore, Billerica, MA, USA). The membranes were incubated successively with primary and secondary antibodies. Antibody binding was detected using Amersham ECL Select Western blotting detection reagent (GE Healthcare, Buckinghamshire, UK). Primary antibodies included light chain 3B (LC3B, 1:1000, Novus Biologicals, Centennial, CO, USA), total extracellular signal-regulated kinase (ERK, 1:1000, Cell Signaling Technologies, Danvers, MA, USA), phospho-ERK (1:1000, Cell Signaling Technologies), autophagy-related 16 like 1 (ATG16L1, 1:1000, Cell Signaling Technologies), and glyceraldehyde-3-phosphate dehydrogenase (GAPDH, 1:2000, Cell Signaling Technologies).

#### 2.2.7. Autophagic Flux Assay

A549 cells were cultured with or without ondansetron, palonosetron, or ramosetron for 24 h. To confirm the autophagic flux, A549 cells were added to Baf-1 (50 nM) 2 h before the harvest for Western blot analysis, washed twice with cold PBS, and lysed in radio-immunoprecipitation assay buffer containing protease inhibitors. Western blot analysis was performed for LC3B and GAPDH antibodies.

#### 2.2.8. Statistical Analysis

All data from the in vitro experiments are expressed as means (standard deviation) and were compared using one-way analyses of variance with Bonferroni post hoc multiple comparisons. A *p*-value <0.05 was considered statistically significant.

## 3. Results

### 3.1. Study Population, Demographic Data, and Perioperative Characteristics

A total of 446 cases of open surgery for lung cancer were reviewed. We excluded 33 cases who died within 30 days after surgery and seven cases diagnosed with stage IV cancer. The data of the remaining 406 patients were examined in this study. During the perioperative period, 308 patients (P-R group) were treated with palonosetron or ramosetron. Meanwhile, 98 patients (No P-R group) did not use these drugs by any method; they received other medications for post-operative nausea and vomiting, such as metoclopramide, ondansetron, or dexamethasone. All patients underwent general anesthesia using volatile anesthetic and remifentanil infusion. For post-operative pain control, epidural patient-controlled analgesia was performed.

Appropriate PSM produced 60 1:2 matched pairs with exclusion of 226 cases (188 cases with palonosetron or ramosetron and 38 cases without palonosetron or ramosetron). Demographic characteristics are listed in Table 1. When comparing demographic data between the two groups after PSM, differences were observed in recurrence and mortality after surgery.

### 3.2. Association between the Use of Palonosetron or Ramosetron, Cancer Recurrence, and Mortality after Surgery

The mean follow-up time in all patients was 40.3 ± 16.1 months. During this time period, cancer recurrence occurred in 72 patients in the unmatched study population. Among the 180 matched individuals, cancer recurrence occurred in 26 (21.67%) patients in the P-R group and 22 (36.67%) patients in the No P-R group (*p* = 0.005). All-cause mortality occurred in 68 patients in the unmatched study population. For the matched population, all-cause mortality occurred in 17 (14.17%) patients in the P-R group and 14 (23.34%) patients in the No P-R group (*p* = 0.004). Kaplan–Meier curves for cancer recurrence and overall mortality of patients treated with and without palonosetron or ramosetron are shown in Figure 1. The log-rank test of recurrence and mortality revealed significant differences between the P-R and No P-R groups (*p* = 0.04 and *p* = 0.116, respectively).

Table 2 shows findings from the Cox regression analysis of factors associated with increasing recurrence rate. According to multivariate Cox regression analysis with 1:2 PSM, the only factor associated with lower recurrence rate was perioperative usage of palonosetron or ramosetron (HR, 0.547; 95% CI 0.308–0.974, *p* = 0.0404). A higher cancer stage (Stage II and III as opposed to Stage I) was found to be associated with a greater risk of recurrence.

Table 3 shows the results of Cox regression analysis for factors associated with increasing overall survival during the follow-up period. According to the multivariate Cox regression analysis with 1:2 PSM, perioperative usage of palonosetron or ramosetron was not associated with lower mortality rate. A higher cancer stage (Stage III as opposed to Stage I) was found to be associated with greater risk of mortality (HR, 5.872; 95% CI 2.179–15.82, *p* = 0.0005).

### 3.3. 5-HT3 Receptor Confirmation in A549 Cell

Flow cytometry experiments were performed to identify 5-HT3 receptor subtypes present in A549 cells. The expression levels of 5-HT3A, 5-HT3B, 5-HT3C, and 5-HT3D + E were 11.6%, 87.5%, 86%, and 87.6%, respectively (results are provided in Appendix A
Figure A1).

### 3.4. 5-HT3RAs Inhibit A549 Cell Proliferation in a Dose-Dependent Manner

To determine the effects of 5-HT3RAs on A549 cells, we performed a cell viability assay using cultures exposed to 5-HT3RAs at varying doses from 0.1 to 40 μg/mL for ondansetron, 0.025 to 8 μg/mL for palonosetron, and 0.01 to 4 μg/mL for ramosetron. In the case of ondansetron, only concentrations greater than 5 μg/mL showed differences in cell viability after the first day, compared to control cells. However, palonosetron (≥0.05 μg/mL) and ramosetron (≥0.1 μg/mL) caused the cells to lose their proliferative capabilities (*p* < 0.05). After the second day, all of the 5-HT3RAs significantly inhibited cell proliferation, compared to control cells, with ondansetron (≥5 μg/mL), palonosetron (≥0.05 μg/mL), and ramosetron (≥0.05 μg/mL) exhibiting the most potent effect (*p* < 0.05) (Figure 2A).

### 3.5. 5-HT3RAs Inhibit Cell Migration and Colony Formation in A549 Cells

We examined the effect of 5-HT3RAs on cell migration and colony formation in A549 cells. Palonosetron (8 μg/mL) and ramosetron (4 μg/mL), but not ondansetron (40 μg/mL), inhibited cell migration, compared to the control (*p* < 0.05) (Figure 2B). The clonogenic assay revealed that ondansetron, palonosetron, and ramosetron treatment all led to a decrease in colony size, compared to the control (*p* < 0.05). However, the sizes of colonies treated with palonosetron and ramosetron were smaller than those treated with ondansetron (Figure 2C).

### 3.6. 5-HT3RAs Induce Autophagy via the ERK Signaling Pathway

When compared with the control cells, the level of LC3 protein was increased in the 5-HT3RA-treated cells, which was accompanied by ERK activation (Figure 3A). When an ERK inhibitor (U0126, 10 mM) was added to the 5-HT3RA-treated cells, LC3 protein levels were reversed (Figure 3B), suggesting that 5-HT3RAs induce autophagy via the ERK pathway in A549 cells. We also investigated whether 5-HT3RAs affect the expression of p62 and ATG proteins, such as ATG3, ATG5, ATG7, and ATG12, which are known to be associated with apoptotic cell death; however, 5-HT3RAs did not change their expression levels, compared to control, in A549 cells (data not shown).

### 3.7. 5-HT3RAs Act as Autophagy Inducers in A549 Cells

The level of LC3 protein was increased in the 5-HT3RA-treated cells compared to the control cells. When Baf-1 was added to the 5-HT3RA-treated cells, the LC3 protein levels were significantly increased compared to the 5-HT3RA-treated cells without the addition of Baf-1 (Figure 4).

## 4. Discussion

5-HT3RAs are widely used for the prevention of post-operative or chemotherapy-induced nausea and vomiting. To the best of our knowledge, this is the first study to investigate the anti-tumor potential of 5-HT3RAs. The findings of the present retrospective analysis suggest the possibility that post-operative administration of palonosetron and ramosetron could lower tumor recurrence after surgery in patients undergoing open thoracotomy for lung cancer. The results of our in vitro experiments also demonstrated that palonosetron and ramosetron inhibit cell proliferation, migration, and colony formation, and that high doses of ondansetron inhibit cell proliferation and migration of A549 cells, which appeared to be associated with autophagic cell death linked to ERK activation.

Serotonin is a well-known monoamine neurotransmitter that is biochemically derived from tryptophan. This small molecule is known to exist mainly in the gastrointestinal tract, blood platelets, and the central nervous system in humans and animals, acting as a gastrointestinal motility controller, growth factor, and anti-depressant. Serotonin is also known to act as a mitogenic factor in normal cells and in cancer cells [15,16,17]. This neurotransmitter acts on various types of 5-HT receptors that comprise a large, complex family of receptors, and promotes the proliferation of various types of cancer cells, including prostate, lung, and colon cancer cells. There are seven known families of 5-HT receptors and 13 receptor subtypes in humans [20]. In particular, 5-HT1 and 5-HT2 receptors were reported to be closely related to the aggressive tendency of cancer cells [15,16,18,19]. Based on these reports, several studies attempted to examine whether antagonism of 5-HT1 and 5-HT2 receptors can prevent cancer cell growth. Unlike the other G-protein-coupled 5-HT receptors, 5-HT3 receptors are involved in pumping sodium and potassium ions during neuron activation [20]. However, it is not known which 5-HT3 receptor subtypes are present in lung cancer cells. From the results of flow cytometry, we found that 5-HT3B, 3C, and 3D + E were present in A549 cells; 5-HT3A was present in small amounts.

In our retrospective analysis, the perioperative use of palonosetron or ramosetron in lung cancer patients was found to decrease post-operative lung cancer recurrence rate. This suggests that 5-HT3 receptors, such as the previously studied 5-HT1 and 5-HT2 receptors, might have the potential to affect tumor mitogenesis. These results are meaningful and indicate that further investigation into the use of 5-HT3RAs as antiemetics in clinical practice is important.

Interestingly, our data indicated that ondansetron does not reduce tumor recurrence or overall mortality after lung cancer surgery. Ondansetron was the first 5-HT3RA developed and possesses a relatively short half-life, compared to other 5-HT3RAs. Ramosetron exhibits a significantly greater binding affinity for 5-HT3 receptors and a slower dissociation rate from receptor binding, resulting in more potent and longer effects (up to 48 h) than ondansetron [21]. Palonosetron is the first of the “second-generation” 5-HT3RAs and has superior effects due to higher receptor binding affinity, compared to first-generation drugs. The mean elimination half-life of palonosetron is about 40 h [22]. Our results demonstrated that palonosetron and ramosetron produce a positive effect on lung cancer recurrence-free survival, whereas ondansetron does not. These effects appear to be related to the potency of 5-HT3RA. Several clinical trials comparing the antiemetic efficacy of 5-HT3RAs also showed that palonosetron and ramosetron are superior to ondansetron [23,24].

Our cell viability experiments were performed with references to serum concentrations of the drugs used in clinical situations, as described in previous studies [25,26,27]. Thus, ondansetron, palonosetron, and ramosetron were tested with 0.1, 0.025, and 0.01 μg/mL minimum concentrations, respectively. The results demonstrated that all three agents inhibited A549 proliferation, although this differed by concentration. After two days of treatment, significant inhibition of A549 cell proliferation was found for palonosetron at two times the actual clinical blood concentration and for ramosetron at five times the actual clinical blood concentration. Meanwhile, ondansetron treatment only inhibited A549 proliferation at concentrations of 50 times or more the clinical blood concentration. These data are consistent with our retrospective results that showed that ondansetron did not decrease recurrence rate or mortality after open thoracotomy for lung cancer. However, a well-controlled prospective study is needed that takes into consideration that these 5-HT3RAs have different durations and frequency of administration.

The role of serotonin on tumor growth is thought to be concentration-dependent [28]. High doses of serotonin exhibit a growth stimulatory effect in aggressive cancers and carcinoids. Results from our in vitro study showed that 5-HT3RAs inhibit proliferation of lung cancer cells in a dose-dependent manner. However, at lower concentrations, palonosetron and ramosetron exhibited greater ability to inhibit lung cancer cell proliferation than ondansetron. Subsequent studies on the ability of 5-HT3RAs to inhibit cancer cells at any dose in vivo are important due to the variation in clinical doses used and binding capacities observed.

The malignant potential of tumor cells is reflected by their increased proliferation, migration, and colony formation abilities. Because these mechanisms directly lead to tumor recurrence, previous studies focused on finding mechanisms of regulation of different types of cancer [29,30]. Our results showed that 5-HT3RAs could impair A549 cell colony formation and that palonosetron and ramosetron could reduce A549 cell migration. Although it was not statistically sufficient, a trend of reduction in migration was also observed in the ondansetron-treated A549 cells. These results suggest that 5-HT3RAs could possess anti-tumor potential in lung cancer cells.

Autophagic cell death, which is another type of programmed cell death, is the process via which the cell degrades its own organelles and some proteins through the formation of autophagosomes, with the nucleus remaining intact until the late stage of cell death [31]. This cellular process is known to play an important role in health and the pathogenesis of many diseases, including cancers [32,33]. Many publications concluded that autophagy stimulates cell death in mammalian cells and non-mammalian systems [34]. Also, ERK signaling was shown to be associated with autophagy by interacting with LC3 [35,36,37]. Our in vitro data show that the expression of autophagic markers, such as LC3, was increased in association with ERK activation (Figure 3A,B).

To confirm the autophagic effect of 5-HT3RAs, we performed an additional experiment using Baf-1. During autophagic cell death, autophagosomes then fuse with lysosomes facilitating the degradation of engulfed cargo by lysosomal proteases. Baf-1, an inhibitor of autophagosome–lysosome fusion, acts by inhibiting the late phase of autophagy [38]. LC3 levels were increased in Baf-1-treated cells compared to the 5-HT3RA-treated cells (Figure 4), demonstrating that 5-HT3RAs act as strong autophagy inducers and the growth inhibitory effects of 5-HT3RAs are associated with autophagy via ERK activation in lung cancer cells.

A previous study in hepatocellular carcinoma showed that inhibition of autophagy is induced via a 5-HT2 receptor related pathway [39]. Research on prostate cancer cells showed that 5-HT1A/B can induce significant growth inhibition and apoptosis [40]. The 5-HT3 receptor acts through mechanisms that are different from other receptors [20], suggesting that the 5-HT3 receptor may function via a unique mechanism for promoting cancer cell survival or growth. Several studies concluded that 5-HT is involved in promoting cell survival and the growth of various cancer cells by activating cancer-specific 5-HT receptors. Thus, although the mechanism is different, this study suggests that 5-HT3RAs can be considered as candidates for cancer therapy. Further research is needed to explore the precise molecular mechanisms underlying this process.

There are potential limitations of this study. Firstly, the main limitation of this study is its retrospective design. Uncontrolled and unrecognized biases are frequent in such studies. Nevertheless, our data can be considered reliable because all patients were managed perioperatively in a similar manner at the same hospital over a study period. Furthermore, anesthetic management was relatively consistent. Secondly, despite planning to study the effects of 5-HT3 on lung cancer, ondansetron, which is a kind of 5-HT3RA, was included in the No P-R group as a control. As mentioned above, the authors considered this to be a problem related to the potency of ondansetron. This limitation of the retrospective study involving low-potency ondansetron should be supplemented with further prospective randomized studies. Thirdly, the patient number in the P-R group was much higher than that in the No P-R group, due to the advantages of drug efficacy and duration. We tried to supplement this limitation with PSM, and we also performed 1:2 matching to obtain sufficient numbers of patients for statistically significant analysis.

Overall, more in vivo research and clinical studies are needed to understand how the mechanisms identified in our in vitro research work together and to determine the actual clinical doses of 5-HTRAs required to have a significant impact on recurrence and survival in patients undergoing surgery for lung cancer.

## 5. Conclusions

The present study showed the possibility that perioperative administration of palonosetron or ramosetron has anti-tumor potential associated with increased recurrence-free survival in patients after open thoracotomy for lung cancer. Our in vitro experiments also showed that 5-HT3RAs suppress the growth of cancer cells via activation of ERK and autophagy. Once additional molecular and prospective clinical studies are conducted to confirm the effects of 5-HT3RA on tumor cells, the anti-tumor potential of palonosetron and ramosetron might be considered when selecting antiemetics in cancer patients, especially during the post-operative period.

## Figures and Tables

**Figure 1 jcm-08-01380-f001:**
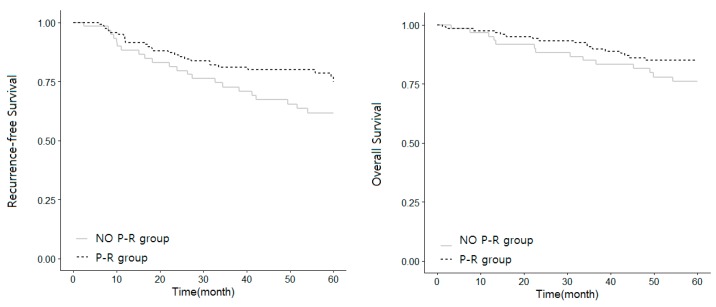
Kaplan–Meier curves for cancer recurrence and overall survival of patients treated with and without palonosetron or ramosetron. P-R group patients were treated with ramosetron or palonosetron perioperatively. No P-R group patients were treated without ramosetron and palonosetron perioperatively.

**Figure 2 jcm-08-01380-f002:**
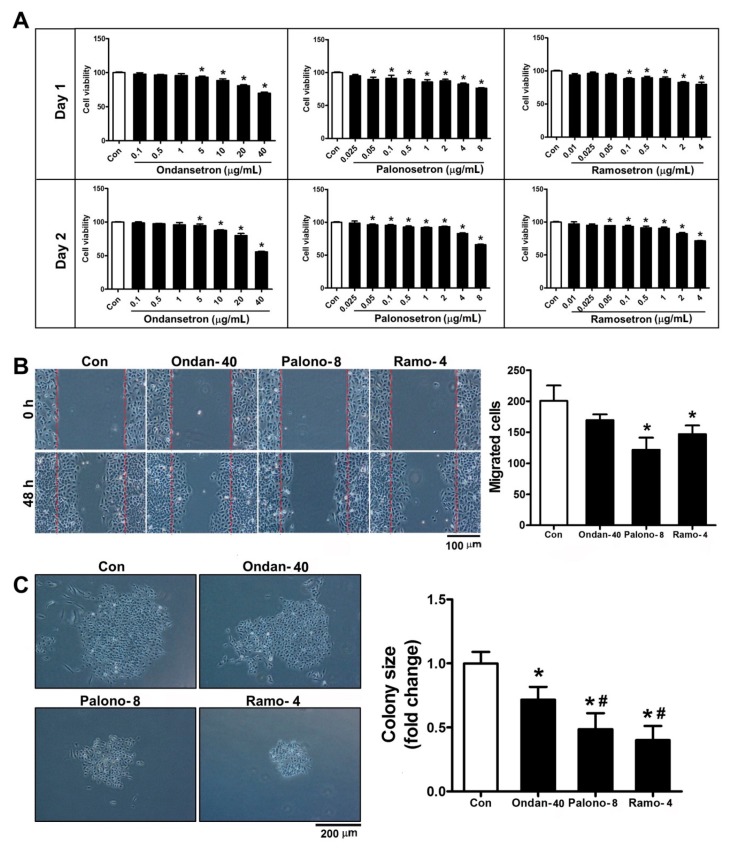
5-Hydroxytryptamine 3 (5-HT3) receptor antagonists inhibit cell proliferation, migration, and colony formation in lung cancer cells. (**A**) Cell viability was measured by EZ-Cytox Cell Viability Assay Kit after one or two days; *n* = 8, * *p* < 0.05 vs. control. (**B)** A549 cells were exposed to ondansetron (40 μg/mL), palonosetron (8 μg/mL), or ramosetron (4 μg/mL) for 48 h. Cell migration was examined with the cell scraping assay. Migrated cells were counted at 48 h post-scrape; *n* = 5, * *p* < 0.05 vs. control (**C**) Colony size was measured using the Image J software program; * *p* < 0.05 vs. control, # *p* < 0.05 vs. ondansetron 40 μg/mL.

**Figure 3 jcm-08-01380-f003:**
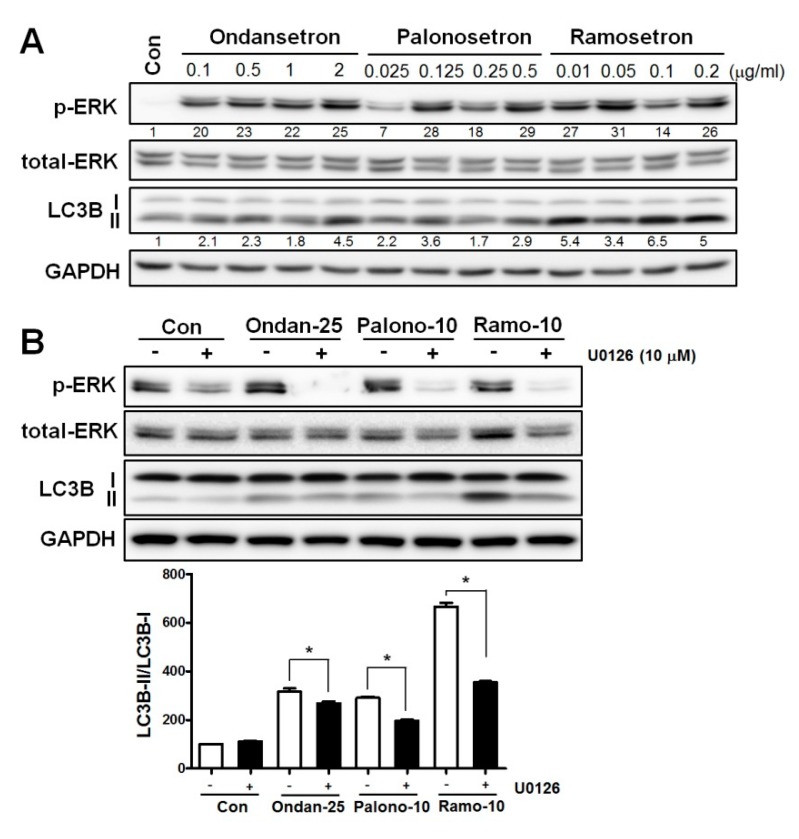
5-HT3 receptor antagonists induce autophagy via extracellular signal-related kinase (ERK) activation in lung cancer cells. (**A**) Levels of phospho-ERK, total-ERK, light chain 3B (LC3B), and autophagy-related 16 like 1 (ATG16L1) were determined by Western blotting. Glyceraldehyde-3-phosphate dehydrogenase (GAPDH) served as a loading control. (**B**) A549 cells were treated with or without ERK inhibitor (U0126, 10 mM) before a 1-h treatment with 5-HT3 receptor antagonist. Levels of phospho-ERK, total-ERK, and LC3B were determined by Western blotting. GAPDH served as a loading control; * *p* < 0.05.

**Figure 4 jcm-08-01380-f004:**
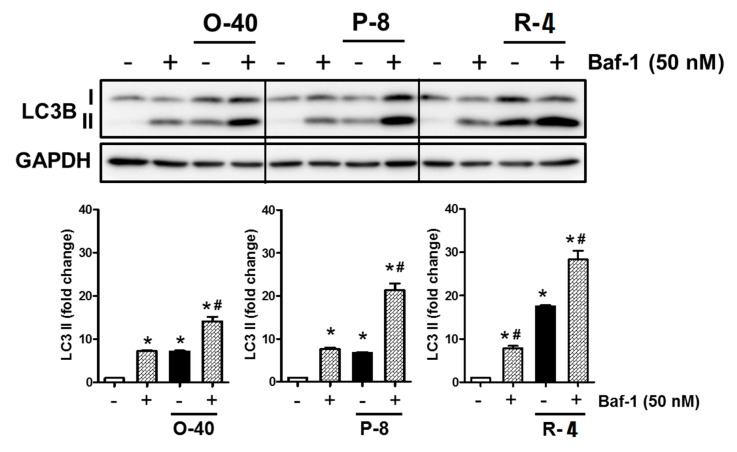
5-HT3 receptor antagonists function as novel autophagy inducers in lung cancer cells. A549 cells were treated with or without ondansetron (40 μg/mL), palonosetron (8 μg/mL), or ramosetron (4 μg/mL) for 24 h and then added to 50 nM bafilomycin A1 (Baf-1) 2 h before observation. The level of LC3B was determined by Western blotting. GAPDH served as a loading control; * *p* < 0.05 vs. untreated cells, # *p* < 0.05 vs. 5-HT3RA-treated cells.

**Table 1 jcm-08-01380-t001:** Demographic and perioperative characteristics.

	Total Population			Propensity-Matched Population		
	P-R Group (*n* = 308)	No P-R Group (*n* = 98)	*p*-Value	P-R Group (*n* = 120)	No P-R Group (*n* = 60)	*p*-Value
Age	68.7 ± 9.7	67.0 ± 10.7	0.144	67.9 ± 9.8	68.4 ± 9.7	0.718
Sex (M/F)	239/69	77/21	0.840	85/35	44/16	0.810
BMI	23.60 ± 2.84	22.93 ± 3.82	0.113	23.73 ± 3.05	23.18 ± 2.99	0.2535
Comorbidity						
HTN	92 (29.87)	35 (35.71)	0.286	34 (28.33)	21 (35)	0.427
DM	43 (13.96)	12 (12.24)	0.666	18 (15)	7 (11.67)	0.377
Cancer stage			<0.0001			0.855
I	228 (74.03)	49 (50)		93 (77.50)	46 (76.67)	
II	39 (12.66)	31 (31.63)		19 (15.83)	10 (16.67)	
III	38 (12.34)	18 (18.36)		8 (6.67)	4 (6.67)	
Pathology			0.1278			0.801
Squamous cell carcinoma	67 (21.75)	30 (30.61)		31 (25.83)	14 (23.34)	
Adenocarcinoma	222 (72.08)	65 (66.33)		84 (70)	44 (73.34)	
Others	19 (6.17)	3 (3.06)		5 (4.17)	2 (3.34)	
Pre-operative CTx	19 (6.17)	9 (9.18)	0.305	9 (7.50)	4 (6.67)	>0.9999
Operative (op) time	179.5 ± 73.3	183.1 ± 77.8	0.702	172.3 ± 71.1	185.3 ± 76.2	0.313
Post-op CTx	65 (21.10)	30 (30.61)	0.053	19 (15.83)	18 (30)	0.077
Post-op RTx	5 (1.62)	2 (2.04)	0.782	2 (1.67)	1 (1.67)	>0.9999
Recurrence	76 (24.68)	44 (44.90)	<0.0001	26 (21.67)	22 (36.67)	0.005
Mortality	64 (20.78)	35 (35.71)	0.003	17 (14.17)	14 (23.34)	0.004

P-R group patients were treated with ramosetron or palonosetron perioperatively. No P-R group patients were treated without ramosetron and palonosetron perioperatively. HTN, hypertension; DM, diabetes mellitus; CTx, chemotherapy; RTx, radiotherapy; M, male; F, female; BMI, body mass index.

**Table 2 jcm-08-01380-t002:** Univariate and multivariate regression analysis of variables after propensity score matching with recurrence after open lung surgery for lung cancer.

Recurrence	Univariate Analysis		Multivariate Analysis	
	Hazard Ratio (95% CI)	*p*-Value	Hazard Ratio (95% CI)	*p*-Value
5-HT3RA usage				
No P-R group	1		1	
P-R group	0.584 (0.331–1.03)	0.0634	0.547 (0.308–0.974)	0.0404
Age	0.987 (0.959–1.015)	0.3682	0.981 (0.954–1.009)	0.1823
Sex				
Male	1			
Female	1.285 (0.705–2.342)	0.4127	1.488 (0.793–2.792)	0.2161
BMI	1.000 (0.91–1.099)	>0.9999		
Cancer stage				
I	1		1	
II	2.534 (1.312–4.895)	0.0056	2.641 (1.357–5.137)	0.0042
III	3.836 (1.671–8.805)	0.0015	4.451 (1.887–10.498)	0.0006
Pre-operative HTN				
No	1			
Yes	0.721 (0.374–1.389)	0.3279		
Pre-operative DM				
No	1			
Yes	1.057 (0.474–2.357)	0.8918		
Pre-operative CTx				
No	1			
Yes	0.78 (0.242–2.513)	0.6777		
Operative time	0.996 (0.992–1.001)	0.0823		
Post-operative CTx				
No	1			
Yes	0.779 (0.377–1.61)	0.5001		
Post-operative RTx				
No	1			
Yes	0	0.9841		
Pathology				
Squamous cell carcinoma	1			
Adenocarcinoma	1.272 (0.613–2.636)	0.5182		
Others	1.626 (0.351–7.525)	0.5342		

P-R group patients were treated with ramosetron or palonosetron perioperatively. No P-R group patients were treated without ramosetron and palonosetron perioperatively. 5-HT3RA, 5-hydroxytryptamine 3 receptor antagonist; HTN, hypertension; DM, diabetes mellitus; CTx, chemotherapy; RTx, radiotherapy; BMI, body mass index; CI, confidential interval.

**Table 3 jcm-08-01380-t003:** Univariate and multivariate regression analysis of variables after propensity score matching with expire rate after open lung surgery for lung cancer.

Expire	Univariate Analysis		Multivariate Analysis	
	Hazard Ratio (95% CI)	*p*-Value	Hazard Ratio (95% CI)	*p*-Value
5-HT3RA usage				
No P-R group	1		1	
P-R group	0.611 (00.301–1.241)	0.1729	0.553 (0.264–1.158)	0.1164
Cancer stage				
I	1		1	
II	2.421 (1.045–5.612)	0.0392	2.085 (0.846–5.139)	0.1106
III	4.5 (1.771–11.436)	0.0016	5.872 (2.179–15.82)	0.0005
Sex				
Male	1			
Female	1.391 (0.666–2.903)	0.3794	1.854 (0.842–4.081)	0.125
Age	0.986 (0.951–1.022)	0.4381	0.974 (0.941–1.009)	0.1402
BMI	1.008 (0.897–1.133)	0.8931		
Pre-operative HTN				
No	1			
Yes	0.398 (0.153–1.037)	0.0594		
Pre-operative DM				
No	1			
Yes	0.628 (0.191–2.065)	0.4434		
Operative time	0.996 (0.99–1.001)	0.0986		
Pre-op CTx				
No	1			
Yes	0.373 (0.051–2.733)	0.3317		
Post-op CTx				
No	1			
Yes	0.517 (0.181–1.478)	0.2184		
Post-op RTx				
No	1			
Yes	0	0.988		
Pathology				
Small cell carcinoma	1		1	
Adenocarcinoma	0.466 (0.218–0.995)	0.0486	0.520 (0.230–1.175)	0.1159
Others	2.097 (0.584–7.521)	0.256	3.565 (0.866–14.681)	0.0784

P-R group patients were treated with ramosetron or palonosetron perioperatively. No P-R group patients were treated without ramosetron and palonosetron perioperatively. 5-HT3RA, 5-hydroxytryptamine 3 receptor antagonist; HTN, hypertension; DM, diabetes mellitus; CTx, chemotherapy; RTx, radiotherapy; BMI, body mass index; CI, confidential interval.

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
