# Peer review of "Anti-Tumor Potential of a 5-HT3 Receptor Antagonist as a Novel Autophagy Inducer in Lung Cancer: A Retrospective Clinical Study with In Vitro Confirmation"

_jcm, 2019, doi:10.3390/jcm8091380_

Round 1
Reviewer 1 Report
Title: Anti-tumour potential of a 5-HT3 receptor antagonist as a novel autophagy inducer in lung cancer: A retrospective clinical study with in vitro confirmation
The authors described the antitumor effect of 5-HT3RA in lung cancer. It is interesting and the manuscript has been well written. However, there are several issues to be addressed before accept.
The reason for selection of A549 lung cancer cell line should be explained. If possible, the data using other type of cell lines would be valuable. Some of the control group, designated as No P-R group, took ondansetron during the perioperative period. Ondansetron is one of the 5-HT3RA group and acted as an autophagy inducer via the ERK signaling pathway in this study, as well as palonosetron or ramosetron. Please show the correctness for the No P-R group of using ondansetron. There some misspelling in the text.
Reviewer 2 Report
The authors instructions have been incorporated into the introduction and materials and methods and should be removed.
A fundamental issue with the patient selection should be addressed: in the NO PR group the patients may have had other antiemetic such as “meteclopramide, dexamethasone OR ondansetron” line 197. It is not clear whether patients in this group have been exposed to 5HT3RA in the form of ondansetron and we are not presented with any data on patients who have had no 5HT3RA at all. This may be because the numbers of available patients are very small (60 in this group), however this means that conclusions cannot be drawn.
The in vitro data demonstrate inhibition of proliferation, migration and clonogenicity with all 5HT3 antagonists albeit at concentrations higher than those used in clinical practice. Ondansetron does demonstrate activity (at higher concentrations) in all 3 assays and it is therefore important to determine whether patients were exposed to this drug in the NO PR group.
Given the uncertainty about the exposure to 5HT3 antagonists in the NO PR group and the small numbers of patients involved, no conclusions can be drawn and certainly no recommendations can be made about prospective use of palenosetron or ramosetron to reduce post operative recurrence of cancer without further studies.
Round 2
Reviewer 1 Report
The authors responded to the comments properly. The manuscript has been revised well.